# *StcU-2* Gene Mutation via CRISPR/Cas9 Leads to Misregulation of Spore-Cyst Formation in *Ascosphaera apis*

**DOI:** 10.3390/microorganisms10102088

**Published:** 2022-10-21

**Authors:** Tessema Aynalem, Lifeng Meng, Awraris Getachew, Jiangli Wu, Huimin Yu, Jing Tan, Nannan Li, Shufa Xu

**Affiliations:** 1Key Laboratory of Pollinating Insect Biology, Ministry of Chinese Agriculture and Rural Affairs, Institute of Apicultural Research, Chinese Academy of Agricultural Sciences, Beijing 100093, China; 2College of Agriculture and Environmental Science, Bahir Dar University, Bahir Dar P.O. Box 26, Ethiopia

**Keywords:** honey bee, *Ascosphaera apis*, CRISPR/Cas9, sporulation, versicolorin reductase gene (*StcU-2*)

## Abstract

*Ascosphaera apis* is the causative agent of honey bee chalkbrood disease, and spores are the only known source of infections. Interference with sporulation is therefore a promising way to manage *A. apis.* The versicolorin reductase gene (*StcU-2*) is a ketoreductase protein related to sporulation and melanin biosynthesis. To study the *StcU-2* gene in ascospore production of *A. apis*, CRISPR/Cas9 was used, and eight hygromycin B antibiotic-resistant transformants incorporating enhanced green fluorescent protein (*EGFP*) were made and analyzed. PCR amplification, gel electrophoresis, and sequence analysis were used for target gene editing analysis and verification. The CRISPR/Cas9 editing successfully knocked out the *StcU-2* gene in *A. apis. StcU-2* mutants had shown albino and non-functional spore-cyst development and lost effective sporulation. In conclusion, editing of *StcU-2* gene has shown direct relation with sporulation and melanin biosynthesis of *A. apis*; this effective sporulation reduction would reduce the spread and pathogenicity of *A. apis* to managed honey bee. To the best of our knowledge, this is the first time CRISPR/Cas9-mediated gene editing has been efficiently performed in *A. apis*, a fungal honey bee brood pathogen, which offers a comprehensive set of procedural references that contributes to *A. apis* gene function studies and consequent control of chalkbrood disease.

## 1. Introduction

Honey bee (*Apis mellifera* L.) broods are subjected to several diseases caused by pathogenic microorganisms. *Ascosphaera apis* (Maassen ex Claussen) (*A. apis*) fungus, which causes chalkbrood disease [1], is one of the most serious fungal diseases affecting honey bees [2]. *Ascosphaera apis* is a common pathogen in honey bee colonies around the world and cause economic losses in apiculture [3,4]. After larvae have ingested spores of *A. apis*, the disease weakens the entire honey bee colony by reducing the number of newly emerged bees, weakening workers, and decreasing honey production [5,6,7,8]. Under certain circumstances, *A. apis* can kill the entire colony [9].

Many techniques and chemicals, both artificial and natural, have been used to manage *A. apis* [10,11,12,13]; however, there is no completely effective way to control the disease [14]. The DNA sequencing has revealed many fungal genome sequences, including those of *A. apis*, revealing previously unknown gene clusters [15,16]. These data provide a basis for further studies on the molecular control of this disease. Genome annotation of *A. apis* has revealed several genes encoding homologs of well-known toxins and the enzymes involved [16,17]. For example, the sterigmatocystin biosynthesis pathway gene, versicolorin reductase gene (*StcU-2*). *StcU-2* (*ver-lA* homolog) is a ketoreductase protein that catalyzes the conversion of versicolorin A (VA) into sterigmatocystin (ST) [18]. ST is a mycotoxin and carcinogenic secondary metabolite produced by *Aspergillus* species [19,20,21].

The *StcU-2* gene is involved in the biosynthesis pathway of ST/aflatoxin (AF)/(polyketides), which are toxic, mutagenic, and carcinogenic natural products [22,23] and are part of mycotoxin biosynthesis in *A. apis* [17]. ST serves as a penutimate intermediate in AF biosynthesis, which is synthesized as an end-product by many ascomycetes [24,25]. The *StcU-2* gene is directly involved in the conversion of VA into ST in AF biosynthesis in *Aspergillus parasiticus* [26]. Thus, disruption of the *StcU-2* gene can block ST production and the accumulation of vercicolorin in *A. paraciticus* [27]. The direct involvement of *StcU-2* in ST biosynthesis in *Aspergillus nidulans* was confirmed by disruption of the *StcU-2* gene [28,29]. In addition, disruption of *StcU-2* in *Aspergillus* also resulted in the accumulation of VA, demonstrating the specific requirement of *StcU-2* for the conversion of VA into ST [30,31]. Coordinating the production of the toxic secondary metabolite ST with sporulation has also been mentioned [30]. AF/ST production is closely tied to sporulation [32]. Therefore, interference of *StcU-2* gene expression may be an effective measure for the control of honey bee chalkbrood disease.

Characterization of cryptic gene clusters in *A. apis* were previously difficult due to the lack of appropriate gene manipulation methods [33,34]. However, the underlying mechanisms affecting *A. apis* virulence to honey bee brood, based on specific targeted gene editing, could aid in honey bee disease control. We developed restricted enzyme mediated integration (REMI)-constructed stable *A. apis* random mutants and studied the in vitro pathogenicity changes to honey bee worker larvae [12]. Notable in vitro pathogenicity differences were confirmed among selected REMI mutants and wild-type samples [35]. Transcriptome analysis of these REMI constructed mutants and the wild *A. apis* strain helped elucidate the mechanism of pathogenicity [34]. However, REMI produces random mutants that are not specific to major pathogenicity genes.

The CRISPR/Cas9 system has been successfully applied in a number of organisms including yeast, fishes, plants, and mammalian cells [36,37,38,39,40,41]. Though CRISPR/Cas9 molecular gene editing systems are well-developed for many model fungal systems [42,43,44,45], it has not yet been developed for the study of *A. apis* gene functions. The genetic engineering tools for *A. apis* are still underdeveloped. A rapid method to accurately manipulate *A. apis* genome and gene expression levels could accelerate the development efforts to improve the control of virulence factor genes and lead to reduced pathogenicity. The main objective of this study is the successful editing and functional evaluation of the *A. apis StcU-2* gene using the CRISPR/Cas9 system.

## 2. Materials and Methods

### 2.1. Fungal Strain, and Culture Conditions

Honey bee larval mummies were collected from Institute of Apicultural Research, Chinese Academy of Agricultural Sciences, soaked in 70% ethanol for 1 min, rinsed three times with sterile distilled water, then inoculated on 20 mL potato infusion dextrose agar (PIDA) medium (50% potato infusion *v*/*v*, 20% D-glucose *w*/*v*, 0.4% yeast extract *w*/*v*, agar powder 1.5% *w*/*v*) base on Pitt fungal medium formula [46], containing 30 μg/mL streptomycin to suppress bacterial growth. The normal mycelial growth and amount of hygromycin B antibiotic level to suppress *A. apis* growth was determined by incubating plates at 30 °C in darkness.

### 2.2. Determining Hygromycin B Antibiotic Resistance Level of Ascosphaera apis

A *hygromycin phosphotransferase* (*hph*) gene was inserted in a single-guide RNA-CRISPR/Cas9 plasmid during *A. apis* gene editing to select for antibiotic (hygromycin B) resistance. Thus, the level of hygromycin B antibiotic completely suppressing wild *A. apis* mycelial growth was first tested by growing 0.5 cm diameter parental wild *A. apis* mycelia on different levels of hygromycin B antibiotic concentrations at 0 µg/mL, 10 µg/mL, 25 µg/mL, 50 µg/mL, 75 µg/mL, and 100 µg/mL. Each level was replicated in 20 PIDA plates following previously published protocols [47,48] and incubated at 30 °C for 7 d. The fungal growth was observed every 12 h, and the growth on different concentrations were compared. The wild *A. apis* mycelial growth limiting hygromycin B concentration level was used for the selection of transformants during CRISPR/Cas9 gene editing.

### 2.3. Screening of the A. apis Target Gene to Be Edited

#### 2.3.1. Primer Designing and Synthesis

To confirm the target gene of *A. apis* and intended genes in the plasmid, different primers were designed using a primer designing tool developed by Primer Quest (https://www.idtdna.com/PrimerQuest/Home/Index, accessed on 25 August 2022) and were synthesized by Sangon Biotech Co. Ltd. (Shanghai, China), (Table 1).

#### 2.3.2. Specific Target Gene Amplification

The specific PCR for the selected target gene was run to check the presence of the targeted gene sequence that matched the protospacer and protospacer adjacent motif (PAM) sequences of the reference genome. PCR amplification was accomplished as follows: 1 μL of the template genomic DNA at 40 ng diluted concentration was added to 9.5 μL double-distilled water, and 12.5 μL T5 PCR mix, and 1 μL of each primer (10 µM working dilution) was mixed. The PCR conditions were held at initial denaturing at 98 °C for 3 min, 30 cycles of 30 s at 98 °C, 30 s at 59 °C, and 30 s at 72 °C. The final extension time was set to 72 °C for 5 min. The DNA template from the untransformed wild strain sample was used as a positive control, template DNA free mix was used as negative control, and the DNA from transformed *A. apis* mutant mycelia was used as experimental material during the PCR amplification reaction. The reaction products were analyzed using gel electrophoresis on a 2% agarose gel. Once the intended gene location was amplified in a specified region, the amplified PCR product was sequenced. The sequence result of *A. apis* wild strains was analyzed compared to the *A. apis* reference genome sequence [16] to determine if any single nucleotide polymorphism affects the designed literal *StcU-2* protospacer and PAM sequences. The sequence result was checked, and the new adjusted protospacer and PAM sequence for the *StcU-2* gene was designed from the first exon region to be synthesized as sgRNA.

### 2.4. Establishment of the CRISPR/Cas9 for A. apis Gene Inactivation

#### 2.4.1. Designing of Experimental sgRNA

To use CRISPR/Cas9 for *A. apis* genome editing, a sgRNA plasmid, enabling expression of the gene of interest, was constructed [49]. The reporter gene of enhanced green fluorescent protein (EGFP), the selectable gene of ampicillin, and the *hph* gene were incorporated to report and screen the successfully transformed plasmids. To design the sgRNA, the experimental target gene site was screened using NGG protospacer adjacent motif (PAM), and the selected sgRNA was ranked for on-target efficiency (Doench et al., 2014) and for off-target effect (Hsu et al., 2013). As a first proof-of-principle of the functionality of the system, the versicolorin reductase gene (*StcU-2*) (AAPI15087) was chosen for encoding gene inactivation.

The 10 best-scoring protospacers, preferably at the first exon and near the start codon of the *StcU-2* gene, were manually curated using BLAST analysis on the *A. apis* reference genome [16]. Then, the three best-scoring protospacers with no predicted off-target actions were selected. A final 20-nucleotide-long protospacer sequence with the 3’-PAM, AGG was chosen at the start of the first exon CDS. The selected protospacer sequence was GTACAGTTTGGCCGCCACAG. The designed PAM and protospacer sequences were synthesized into an all-in-one circular sgRNA plasmid, pFungi-argB promotor-hph-terminator-tef1 promotor-NLS-Cas9-Linker-EGFP-NLS-HH ribozyme-sgRNA-backbone-HDV-ribozyme-terminator, which contained the *Aspergillus* nodulous codon-optimized Cas9, *EGFP*, and hygromycin B phosphotransferase (*hph*) coding sequences (SyngenTech Laboratory of Synthetic Biology, Beijing, China) (Figure 1).

The sgRNA sequence was flanked by two ribozyme sequences: hammerhead (HH) on the 5’-end and hepatitis delta virus (HDV) at the 3’-end. The ribozyme sequences ensured the liberation of the sgRNA from the transcript in the nucleus. The expression of the sgRNA containing transcript was controlled by the strong constitutive *A. nidulans* argB promoter and the tef1 terminator. The circular plasmid was transformed into *A. apis* genes via protoplast transformation. The transfected plasmid was self-replicating and did not require integration into the genome. This allowed for easier isolation of cured mutants after growth on non-selective media. The resulting mutants were hygromycin sensitive and could be transformed again using the same screen gene.

#### 2.4.2. Protoplast Isolation and Transformation of *A**. apis*

Protoplast isolation was conducted following Wubie 2014, with slight modification [50]. Actively growing mycelia from solid PIDA media were transferred into 70 mL of potato infusion dextrose liquid media (PIDB) containing 30 μg/mL streptomycin at 30 °C and 140 rpm. The growing mycelia were harvested by filtering through a 47 μm nylon filter, washed with 0.7 M NaCl solution and digested in a driselase lysing enzyme (Sigma Aldrich Chemicals, Beijing, China) dissolved in 0.8 M citric acid monohydrate: NaCl solution (50:50 *v*/*v*). The mycelia–driselase suspension containing 20 mg/mL was shaken at 90 rpm and 30 °C for 3 h to remove the fungal cell wall.

Protoplasts from the lysed mycelia–protoplast suspension were recovered through filtering two loose layers of 30 μm nylon filter, diluted with an equal volume of STC Solution (1 M/L sorbitol, 100 mM/L CaCl_2_2H_2_O, 50 mM/L of 1 M/L Tris-HCl, pH 7.4), and precipitated at 2800 g for 5 min at 4 °C. The driselase solution was discarded, and the pelleted protoplasts were again washed two or three times with 10 mL STC buffer solution at 2800 g for 5 min at 4 °C by discarding the supernatant. Finally, 1 mL of STC buffer solution was added to the pelleted protoplasts as stabilizer. Protoplast counts per mL were checked using hemocytometer (Beyotime Institute of Biotechnology, Beijing, China) to the final concentration of 5 × 10^7^ protoplasts/mL. The isolated *A. apis* protoplasts were used as recipients for transformation. All the transformation steps were completed under aseptic conditions.

For CRISPR/Cas9 gene editing, about 100 μL of protoplasts (10^6^) were added to 2 mL test tubes. About 10 μg DNA of all-in-one circular plasmid sgRNA diluted in sterile distilled water was added and vortexed 10 times for 1 s each, and 50 µL of 40% (*w*/*v*) PEG 4000 solution was added to the protoplast–plasmid mixture and vortexed 5 times 1 s each, followed by 30 min of chilling on ice. DNA uptake was induced with PEG as it clumps and fuses protoplasts, facilitating the trapping of DNA to the fungal protoplasts. The vector–protoplast mixture was then diluted dropwise with an additional 1 mL of 40% PEG 4000 at about 5 cm high. The mixture was gently swirled and incubated at room temperature for another 15 min. After 15 min, the volume was increased to 1.75 mL with STC. The PEG-treated protoplast suspension in the 2 mL test tubes was then centrifuged at 2880× *g* for 10 min at 4 °C, and the supernatant was discarded. About 500 μL of LRM (1 M sorbitol, 0.2% yeast extract *w*/*v*, 0.2% Tryptone *w*/*v*) was added to the pelleted protoplast for cell wall recovery and incubated in a dark 25 °C incubator overnight while shaking at 70 rpm.

#### 2.4.3. Screening of Target Gene Edited *A**. apis* Mutants, Mating Test, and Sporulation Verification

After 24 h of cell wall recovery, the whole protoplast was inoculated onto solid regeneration agar medium (1 M sorbitol, 0.2% yeast extract *w*/*v*, 0.2% Tryptone *w*/*v*, 0.9% agar *w*/*v*) in sterile distilled water containing a concentration of 25 μg/mL hygromycin B antibiotic to suppress the growth of non-transformed colonies and incubated for another 24 h at 30 °C. After 24 h of regeneration in a dark incubator, another thin layer of same solid regeneration agar medium furnished with 25 μg/mL hygromycin B was overlaid and incubated at 30 °C until effectively transformed protoplasts were grown and penetrated the overlay. A 30 μg/mL streptomycin treatment was applied to all solid regeneration media to suppress bacterial growth. Then, 20 rapidly growing hygromycin B-resistant single colonies were chosen and transferred to PIDA agar containing 25 μg/mL hygromycin B for further isolation and analysis. The continuously and rapidly growing colonies were maintained on selection medium for more than five consecutive screenings. They were then isolated by expression of EGFP in the vector DNA construct using a laser scanning microscope. A confocal laser scanning microscope (CLSM) TCS SP2 (Leica Microsystems GmbH, Wetzlar, Germany) was used for microscopic observation of mycelial EGFP. EGFP excitation was achieved with 488 nm argon krypton laser emissions, and observations were made using a 0.8 numerical aperture 20× oil immersion lens. Transformants were microscopically detected and screened at the early mycelial growth stage. For analysis of EGFP expression, individual hygromycin B-resistant and *EGFP*-marker-positive colonies were analyzed using PCR amplification to check *EGFP*, *hph* gene, and target gene site of genomic DNA for the mutants and wild-strain alleles. Amplified target gene PCR products of putative transformants and wild-strain were analyzed using gel electrophoresis and sequenced to verify gene editing [51]. Multiple sequence alignment analysis was conducted using NCBI and UGENE-Clustal W software [52,53].

Sporulation tests between different Cas9-edited *A. apis* mutants were carried out by growing multiple mutant cultures of *StcU-2* in single PIDA Petri dishes between the two mutant inocula and checked for the presence of a thick black spore barrage formed at the interface of the opposite mating-type colonies in a single Petri dish.

## 3. Results

### 3.1. Determination of Hygromycin B Resistance

The minimum *A. apis* growth limiting concentration of hygromycin B for CRISPR/Cas9-based gene editing using *hph* gene-incorporated plasmids was 25 µg/mL (Figure 2a). The mycelial growth was highly limited, and the initially inoculated 0.5 cm in diameter mycelia slowly regressed within 4–5 d. The concentration level was determined after testing multiple hygromycin B concentrations. On the same medium, the control (hygromycin B furnishing was zero) wild-strain mycelial inocula were well grown and sporulated (Figure 2b). Hygromycin B, at a concentration of 25 μg/mL, was adequate to restrict the growth of wild *A. apis* mycelia and function in transformant selection (Figure 2c,d).

### 3.2. Laser Scanning Microscopic Analysis of CRISPR/Cas9-Edited A. apis Mutant Colonies

The total CRISPR/Cas9-gene-edited mutant development efficiency in *A.*
*apis*, based on hygromycin B antibiotic-resistant colonies, was 0.011% per 10^6^ protoplasts. *EGFP* expression of *A. apis* transformants was confirmed by generating emission spectrum profiles (CLSM lambda scan, wavelengths: 500–530 nm) for *StcU-2* mutants (Figure 3a,b) compared to the wild strain group (Figure 3c,d).

### 3.3. PCR and Sequence Analysis of Transformants

All eight CRISPR/Cas9-gene-edited *A. apis* mutants for the *StcU-2* gene gave specific bands of *EGFP* (Figure 4a) and *hph* genes (Figure 4b), except for the wild-strain DNA templates and the negative control (−Ve). The detection of bands for *EGFP* and *hph* genes from the PCR products of all eight hygromycin B antibiotic-resistant transformants of *StcU-2* genes confirmed the proper insertion of the all-in-one plasmid inside the transformants’ nucleus. As a result, proper all-in-one plasmid gene insertion into the nucleus of protoplasts indicated *A. apis* gene editing. The PCR amplification of *EGFP* and *hph* genes confirmed successful insertion of the *EGFP* and *hph* gene carrier all-in-one plasmid into the nucleus of *A. apis* transformed mutants.

The target gene of successful individual transformant colonies, resistant to hygromycin B gold, expressing *EGFP*, and incorporating *hph* gene, was PCR amplified to verify the insertion or deletion of specific nucleotides. Primers designed for the screening of the initial protospacer incorporating target site were used for the verification of correct chromosomal mutations during the mutants’ genomic DNA PCR amplification. The anticipated specific *StcU-2* gene band size for wild-strain *A. apis* colonies was 613 bp. During PCR amplification of effective mutants, specific and nonspecific bands were identified due to the loss and insertion of nucleotides. For all effective mutants, the resulting band sizes showed differences compared with the expected band size of the wild-strain *A. apis* (Figure 4c).

The *EGFP* expression, *hph* gene insertion, and Cas9-induced, targeted-gene-specific site mutations were verified through sequence analysis. The PCR product sequencing results of the *EGFP* gene for different mutants’ alignment showed the same sequence as the anticipated plasmid *EGFP* reference gene sequence.

The PCR products amplified from the putative target-gene-edited mutants were then sequenced (Biomed, Beijing, China) and analyzed for sequence matching using NCBI and Clustal W sequence alignments. Mutant and wild-strain sequence mismatches at targeted-gene-editing site sequences were considered as indicators of successful CRISPR/Cas9 gene editing. Thus, the analysis of the target gene sequence results showed that transformed *A. apis* mutants were successfully generated via CRISPR/Cas9 gene editing.

Targeted gene-editing sequences between eight different *A. apis* Cas9 mutants for the selected genome site showed variation in individual NHEJ effects and consequent gene editing. Bi-allelic, tri-allelic, and multiple loci deletions were visualized among different *StcU-2* mutants’ genome sequences (Figure 5). The sequence analysis of the corresponding PCR fragments revealed deletions or replacements of 1 to 613 base pairs for *StcU-2* mutants. Due to the larger targeted gene section deletion events in some mutants, there were no PCR-amplified bands compared to the wild *A. apis* strain.

### 3.4. Growth Observation and Sporulation of Mutants

We observed the growth and sporulation characteristics of each selected *StcU-2*-edited *A. apis* mutant. Compared to the wild-strain of the fungus, the mycelial growth of the selected *StcU-2* edited *A. apis* was active and fluffy (Figure 6a). The spore-cyst production characteristic of mutants was also greatly reduced (Figure 6b). When observed using a 60×light microscope, the mutants showed high number of regressed albino spore-cysts, all under-developed, unlike the large spore-cyst balls of the control wild-strain *A. apis* colonies (Figure 6c).

Mating type compatibility tests among different Cas9-edited *A. apis StcU-2* mutants were conducted by growing eight mutant cultures in a single (PIDA) plate with a distance of 3 cm between the two inocula and checking for the presence of a thick black spore barrage formed at the interface of the opposite mating type colonies. The edited mutants did not form a black spore-cyst barrage at their interface lines (Figure 6d).

The same mating type compatibility test between different *A. apis* wild-strain groups was conducted by growing multiple wild *A. apis* strain cultures in single PIDA plates and checking for the presence of a thick black spore barrage at the interface of the opposite mating type colonies. This test produced multidirectional black spore-cyst lines at the junctions (Figure 6e), indicating effective mating and sporulation.

## 4. Discussion

Fungi are common pathogens of insects. *A. apis* causes chalkbrood disease and is a major pathogen of the honey bee. Its incidence and severity are increasing due to global climate change [11,54]. Many treatments, including fungicides [55], plant essential oils [56], and biological strategies [13,57], have been used to control chalkbrood disease. However, some of these approaches are not healthy for honey bees and contaminate the wax and honey. There are no effective measures to reliably control chalkbrood. Exploring effective and bee-friendly chalkbrood controlling approaches remain significant for apiculture and the environment. Spores are the primary infective agents of chalkbrood disease. Therefore, interference with spore production may help to control the disease. We used CRISPR/Cas9 gene-editing approaches and found that the *StcU-2* gene functions in the sporulation and melanin biosynthesis in *A. apis*. Mutations of *StcU-2* gene have substantial effects on spore-cyst production and can greatly reduce spore production. These results indicate that interfering with *StcU-2* gene expression can interrupt sporulation in *A. apis* and that CRISPR/Cas9 can be successfully used in *A. apis* gene editing. This study provides baseline data for genes engineered using the Cas9/sgRNA-mediated method in *A. apis*. The results increase understanding of the function and mechanisms of different *A. apis* genes.

### 4.1. Successful Mutation of the StcU-2 Gene Inhibits A. apis Sporulation and Melanin Production, Which Provides an Effective Way to Control Diseases Caused by Ascosphaera

Although *A. apis* alone is rarely fatal for honey bee colonies, it can kill colonies that are also interacting with other fungi [58,59,60]. Additionally, *Ascosphaera* contains 28 species, which are all specialized to infect bees or bee colonies, The susceptibility of honey bees and other economic bee species to *Ascosphaera* varies [61,62,63]. Outbreaks of chalkbrood in the alfalfa leafcutting bee (*Megachile rotundata*) caused by *A. aggregata* can cause great economic losses [64]. Hence, an effective way to prevent chalkbrood disease will benefit *A. apis* and the control of other *Ascosphaera.* The mechanism used by *A. apis* to be a successful entomopathogenic fungus depends on invasion and colonization strategies based on the production of an excessive number of spores. Spores germinate and survive after being ingested by larvae [65]. Cadavers of infected hosts are used to optimize the production and dispersion of asco-spores [66], which are the unique infective stage [50]. To cause pathogenicity, the population of asco-spores in the gut of the host should be 5 × 10^5^ asco-spores/larva [3] and kill the host (larvae) [66]. Thus, a reduction in the quality and number of spores of *A. apis* will be related to reduced spread and a consequent reduction in *A. apis*’s pathogenicity to managed honey bees. In this study, *A. apis* mutants of the *StcU-2* gene caused effective spore-cyst production loss and inhibited spore production in laboratory culture. These results are in line with the finding that decreasing the spore formation reduces the infection of *Verticillium dahliae* [67] and *Aspergillus species* [68].

Melanin of some fungi is spore extracellular matrices or part of the spore wall, and they can help the pathogenic fungi to invade, trick the host recognizing antigen, and block their immune system [69,70]. In *A. apis*, melanin is compacted as an electrodense layer throughout the entire cell wall of the spore and provides the necessary force to promote the mycelium’s penetration of the larvae’s gut wall barrier during the invasion process [71]. Additionally, *A. apis*’s melanin might help spores survive under adverse environments and enable *A. apis* to survive and remain infectious after mummy larvae have been dead for a long time [72]. In this study, all stable *A. apis StcU-2* mutants grown from single protoplasts showed the expected phenotype of the colonies. They had albino spore-cysts that lacked melanin production, indicating that successive inactivation of the *StcU-2* gene was highly related to melanin biosynthesis. Therefore, the successful inhibition of melanin of *A. apis* via interrupting the *StcU-2* gene will reduce the *A. apis*’s pathogenicity.

### 4.2. CRISPR/Cas9 Can Be Used for A. apis Gene Function Studies

The recent advancements in sequencing technology have revealed a number of previously unknown genes of *A. apis* [16,73]. Characterizations of these cryptic genes’ functions in *A. apis* were greatly hampered due to the lack of handy gene manipulation methods. CRISPR/Cas9-mediated gene editing is a powerful tool for uncovering gene functions [74]. However, there is no record for CRISPR/Cas9 used in *A. apis. A. apis*-targeted gene disruptions using CRISPR/Cas9 nuclease at a specific genomic location induced effective DNA double-strand breaks, which were then repaired via the error-prone NHEJ DNA repair pathway. This resulted in insertions and/or deletions (indels) of nucleotides, which then changed gene function [75]. An early exon DSB would easily be repaired via NHEJ repair, causing indels or resulting in a frame shift which will ultimately lead to a premature stop codon or will most likely lead to no protein product being translated at all [76]. Even though a truncated protein is produced, it will be non-functional, as the sequence is altered at an early stage [77]. Thus, the first coding exon region intended sgRNA of *A. apis* produced successful cleavage of the *StcU-2* gene with varied nucleotide indels and phenotypic and functional changes in sporulation. These results demonstrated that CRISPR/Cas9-mediated gene editing of the *A. apis* genome shows the potential of using CRISPR/Cas9 in different *A. apis* gene function studies. In this study, we edited components of molecular pathways likely to contribute to melanin biosynthesis (sporulation) pathways. In comparison to traditional gene deletion approaches, the CRISPR/Cas9 mutagenesis approach for *A. apis* was advantageous, as the system remained active and successively mutagenized the target genes. Hence, the method provides the opportunity to retransform the cured strain using the same selection marker to obtain double-mutant strains.

### 4.3. Minimum Growth-Limiting Level of Hygromycin B Antibiotic to Wild-Strain A. apis Colonies Was Determined

The minimum growth-limiting level of hygromycin B antibiotic to wild-strain *A. apis* colonies, which was critical for the selection of transformant colonies after CRISPR/Cas9 gene editing, was 25 µg/mL. The previous hygromycin B concentration level in PDA media was suggested to be 50 µg/mL [35,50], but there was no spore or mycelial growth observed in the wild-strain *A. apis* in the referred media in this study. The protocol helps to identify *A. apis* transformants from non-transformed colonies. Hygromycin B-resistant *A. apis* colonies can easily be identified by their ability to grow on PIDA plates containing 25 µg/mL hygromycin B. At hygromycin B levels below 25 µg/mL, some colonies have restricted growth, but at 25 µg/mL, all wild-strain *A. apis* colonies were completely inhibited and regressed. This level was used for CRISPR/Cas9 transformants, which incorporated the *hph* gene in the Cas9 enzyme carrier all-in-one plasmids. The contradiction with previous studies regarding hygromycin B [50], might be due to the temperature of the PDA media during the introduction of the hygromycin B antibiotic, which could denature the antibiotic. Though there are thermo-resistant hygromycin B antibiotic strains that resist denaturing above 60 °C [78], the ordinary hygromycin B antibiotic denatures at temperatures above 50 °C [79]. In this study, the hygromycin B antibiotic was added when the medium’s internal temperature was 48 °C.

## 5. Conclusions

We, for the first time, have demonstrated effective CRISPR/Cas9-mediated *StcU-2* gene editing in *A. apis* and confirmed that the *StcU-2* gene was related to sporulation and melanin biosynthesis. This result provides a promising way to reduce pathogenicity and the spread of *A. apis* though CRISPR/Cas9.

## Figures and Tables

**Figure 1 microorganisms-10-02088-f001:**
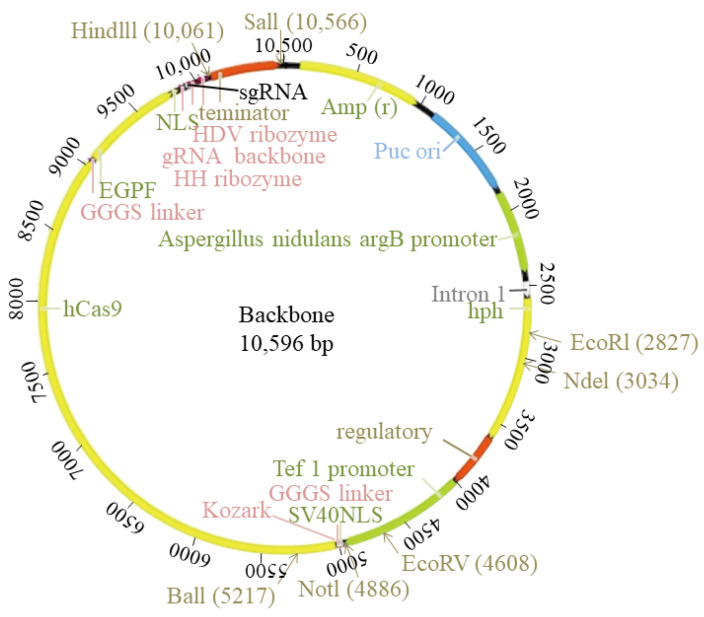
CRISPR-Cas9-based gene disruption all-in-one circular plasmid.

**Figure 2 microorganisms-10-02088-f002:**
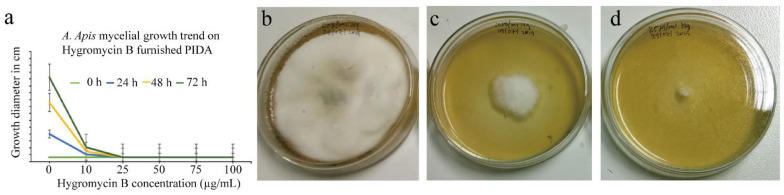
*Ascosphaera apis* (*A. apis*) mycelial growth limiting concentration level of hygromycin B. (**a**) Graphical hygromycin B resistance level test of *A. apis*; (**b**) 0 µg/mL hygromycin B; (**c**) 10 µg/mL hygromycin B; (**d**) 25 µg/mL hygromycin B.

**Figure 3 microorganisms-10-02088-f003:**
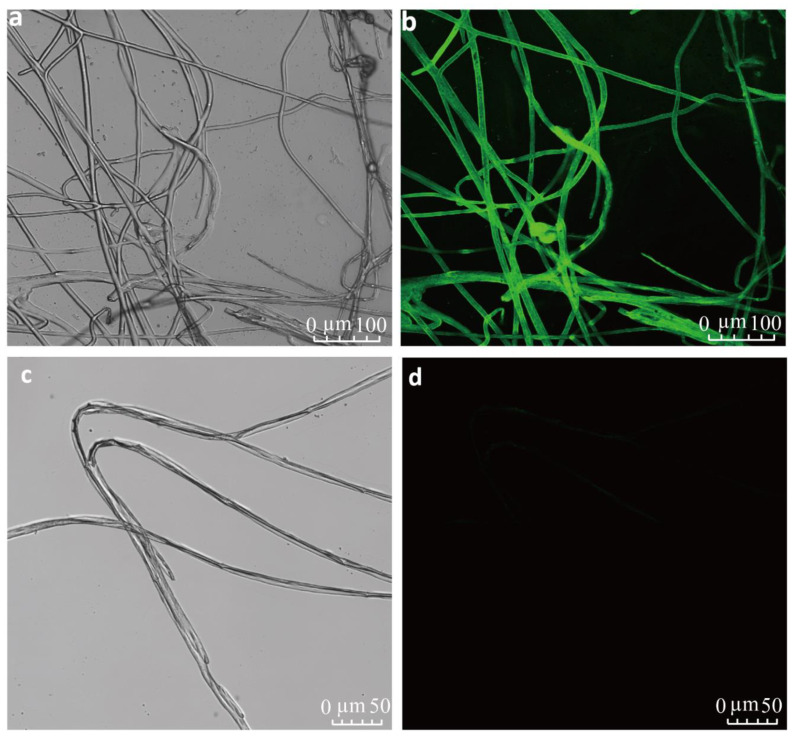
Expression of *EGFP* in transformed *Ascosphaera apis* (*A. apis*) mutant mycelia with plasmid containing *EGFP* and wild-strain group viewed using a confocal laser scanning microscope illuminated with a UV light. (**a**,**b**) Mycelia of transformed *A. apis StcU-2* mutants expressing *EGFP*. (**c**,**d**) Mycelia of *A. apis* wild strain (non-transformed) without expressing *EGFP*.

**Figure 4 microorganisms-10-02088-f004:**
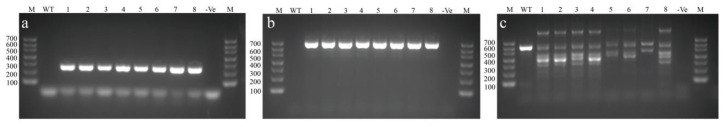
Different amplified *StcU-2*-gene-edited *Ascosphaera apis* (*A. apis*) target gene bands. (**a**) Bands of wild-strain, eight stably selected Cas9-gene-edited *StcU-2* mutants of *A. apis*, and negative control for *EGFP* gene integration. (**b**) Bands of wild-strain, eight stably selected Cas9-gene-edited *StcU-2* mutants of *A. apis*, and negative control for *hph* gene integration. (**c**) Bands of wild-strain, eight stably selected Cas9-gene-edited *StcU-2* mutants of *A. apis*, and negative control for *StcU-2* gene being edited. For all: M denotes the DNA Marker 700 ladder; WT denotes *A. apis* wild strain. Lanes 1–8 denote different independent *StcU-2* transformant mutants; AAStcU-2-1911-1, AAStcU-2-1911-2, AAStcU-2-1911-3, AAStcU-2-1911-4, AAStcU-2-1911-5, AAStcU-2-1911-6, AAStcU-2-1911-7, and AAStcU-2-1911-8. –Ve denotes the negative control, which is a mix of sterile, double-distilled water and PCR without a template.

**Figure 5 microorganisms-10-02088-f005:**
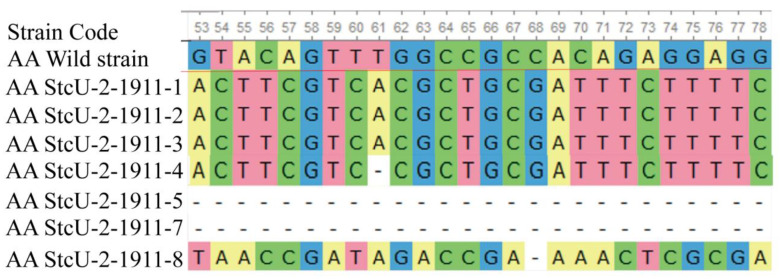
Clustal W sequence alignment of *StcU-2* gene-edited mutants. AA Wild-Strain denotes *Ascosphaera apis* (*A. apis*) wild-strain protospacer sequence and PAM sequence. AAStcU-2-1911-1, AAStcU-2-1911-2, AAStcU-2-1911-3 AAStcU-2-1911-4, AAStcU-2-1911-5, AAStcU-2-1911-7, and AAStcU-2-1911-8 denote different independent *StcU-2* gene-edited mutant protospacer sequences and PAM sequence site editing.

**Figure 6 microorganisms-10-02088-f006:**
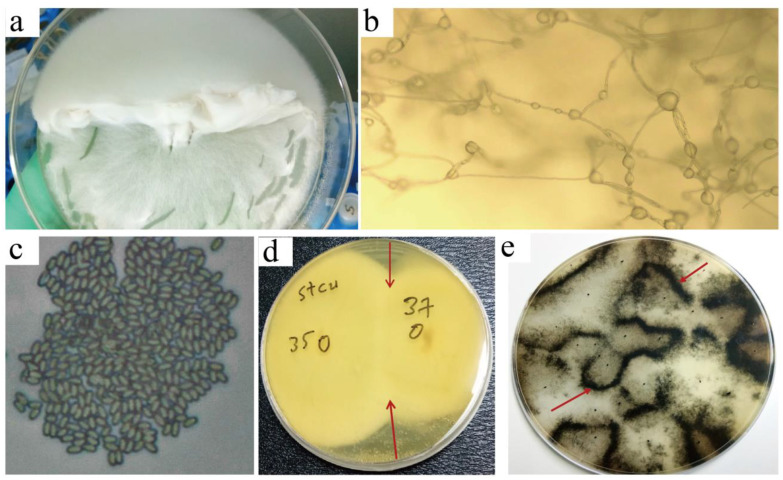
Mating type compatibility test and effective sporulation of different *Ascosphaera apis* (*A. apis*) strain. (**a**) Sporeless fluffy mycelial growth structure of *A. apis* mutant colonies. (**b**) Matured and remarkably small sized *A*. *apis* mutants spore cysts. (**c**) Effective matured *A*. *apis* wild strain asco-spores. (**d**) Mating type compatibility and asco-spore development test among different Cas9 edited *StcU-2* mutant *A*. *apis* colonies. Arrows show line of spore cyst barrage for *StcU-2* mutants, and wild strain *A*. *apis* after mating. Numbers denote individual mutants. Black dots denote initial mycelial inoculation sites. (**e**) Effective mating and sporulation of wild strain. Arrows show line of spore-cyst barrage.

**Table 1 microorganisms-10-02088-t001:** List of primers to amplify targeted gene.

Type of Gene	Forward Primer (F)	Reverse Primer (R)	Band Size
*EGFP*	GCTGACCCTGAAGTTCATCTG	CACCTTGATGCCGTTCTTCT	367 bp
*hph*	GATATGTCCTGCGGGTAAA	CCGTCAACCAAGCTCTGATA	710 bp
*StcU-2*	GTCGGCTGTCAAGTTCCTCA	AGACCCTGACGCAGAACAAG	613 bp

## Data Availability

Not applicable.

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
