# Peer review of "StcU-2 Gene Mutation via CRISPR/Cas9 Leads to Misregulation of Spore-Cyst Formation in Ascosphaera apis"

_microorganisms, 2022, doi:10.3390/microorganisms10102088_

Round 1
Reviewer 1 Report
This is a very clearly written study addressing an important question. The data are mostly clear and are properly interpreted by the authors. My only concerns are mostly small issues with the text, and apparent mislabeling of some panels in the figures.
Line 17, 66, 222, 292, 294, 305, 343, 385, 399: “edition” should be “editing”
Line 46: please define “ST” as “sterigmatocystin”
Line 50: “StcU-2 is the penultimate intermediate”: this seems incorrect. Do the authors mean “ST is the penultimate intermediate”?
Line 66: Please define “REMI” as “restricted enzyme mediated integration”.
Line 134, 212: EGFP is listed as a “selectable marker gene” but it is not strictly possible, as far as I know, to “select” for GFP expression. One can “screen” for it or “sort” for it or “isolate” on the basis of it, but there is no condition of which I’m aware where cells expressing GFP live and the rest die, which is the common definition of “selection” in genetical analysis.
Line 207, 209: I suggest replacing “aggressively” with “rapidly”
Lines 214, 215: “EGFP” should not be italicized, since it’s referring to the gene product
Line 216: “0.8 20X oil immersion lens”: what does the “0.8” refer to here? Numerical aperture?
Figure 2: I am confused about what is shown in panel “d”. The body test suggests that this is the control inocula grown and sporulated on medium lacking hygromycin, and that is what the image looks like to me. However, The figure legend suggests that the medium contains 25 µg/mL hygromycin. Please clarify.
Line 244: Should the “106” here instead be 10^6, that is, with the “6” superscripted, equivalent to 1 million?
Figure 4: the legend does not specify for panel “c” what gene was amplified. Presumably this is StcU-2?
I do not think the sequence of the EGFP gene shown in Figure 5 is worth including as a main figure. Obviously EGFP is present, or else the cells would not be fluorescent; the sequence of the gene provides no new information here.
Line 299: “Due to larger targeted gene section deletion events in some mutants, there were
no PCR amplified bands compared to the wild A. apis strain.” What PCR product did the authors sequence to generate the data shown in Figure 6? The figure legend should specify this information. I am unsure what the absence of amplified bands refers to. In Figure 4c, there seemed to be PCR products amplified in all of the mutants. Assuming that panel represents PCR reactions targeting StcU-2, how do the authors explain the lack of PCR amplified bands that they refer to?
The panels in Figure 7 do not match the figure legend.
Line 390: “The self-replicating plasmids were lost once the selection pressure was removed.” What is the evidence that this is the case?
Author Response
Dear Dr. Calvete and two respected reviewers,
We sincerely appreciate the opportunity to revise our submission. The comments and suggestions have greatly helped us to improve the quality of our manuscript. We have now carefully considered each of the comments and addressed all in the revised manuscript, and checked all the references. Please see our point-by-point responses below (in red).
Response to Reviewer 1 Comments:
Reviewer #1:
Point 1. This is a very clearly written study addressing an important question. The data are mostly clear and are properly interpreted by the authors. My only concerns are mostly small issues with the text, and apparent mislabeling of some panels in the figures.
Response 1: Thank you for your comments and suggestions. We have double-checked the text and the figures, and corrected the mistakes involved in the manuscript.
Point 2. Line 17, 66, 222, 292, 294, 305, 343, 385, 399: “edition” should be “editing”
Response 2: Thank you for your suggestions. All the “edition” which in Line 17, 66, 222, 292, 294, 305, 343, 385, and 399 have been revised as “editing”.
Point 3. Line 46: please define “ST” as “sterigmatocystin”
Response 3: Thank you for your suggestions. The “ST” in Line 46 has been defined as “sterigmatocystin”.
Point 4. Line 50: “StcU-2 is the penultimate intermediate”: this seems incorrect. Do the authors mean “ST is the penultimate intermediate”?
Response 4: Thank you for your comments. Yes, here in the text, “ST” is the penultimate intermediate. We have revised “StcU-2 is the penultimate intermediate in the 50 functionally conserved aflatoxin pathway and is synthesized as an end product by many 51 ascomycetes” as “ST serves as a penutimate intermediate in aflatoxin biosynthesis, which is synthesized as an end product by many 51 ascomycetes” from Line 50 to 51.
Point 5. Line 66: Please define “REMI” as “restricted enzyme mediated integration”.
Response 5: Thank you very much for your suggestion. We have corrected as suggested in Line 66.
Point 6. Line 134, 212: EGFP is listed as a “selectable marker gene” but it is not strictly possible, as far as I know, to “select” for GFP expression. One can “screen” for it or “sort” for it or “isolate” on the basis of it, but there is no condition of which I’m aware where cells expressing GFP live and the rest die, which is the common definition of “selection” in genetical analysis.
Response 6: We are grateful for your quite academic comments. Yes, the GFP we used in this experimentation is as a reporter gene. According your comment, we have corrected the mistake in Line 134, the sentence has been revised as: “the reporter gene of enhanced green fluorescent protein (EGFP) the selectable gene of ampicillin and hph gene were incorporated to report and screen the successfully transformed plasmids.”
In Line 212 has been revised as “They were then isolated by expression of Enhanced Green Fluorescent Protein (EGFP) in the vector DNA constructed using a laser scanning microscope”.
Point 7. Line 207, 209: I suggest replacing “aggressively” with “rapidly”
Response 7: Thank you for your suggestion. We have revised “aggressively” with “rapidly” as suggested in Line 207 and Line 209.
Point 8. Lines 214, 215: “EGFP” should not be italicized, since it’s referring to the gene product
Response 8: Thank you for your suggestion. “EGFP” has been corrected to non-italic type in Line 214 and 215.
Point 9. Line 216: “0.8 20X oil immersion lens”: what does the “0.8” refer to here? Numerical aperture?
Response 9: We are very sorry for our bewilder expression. Yes, “0.8” here refers to the numerical aperture is 0.8. Based on your suggestion, we have revised the “0.8 20X oil immersion lens” as “0.8 numerical aperture 20× oil immersion lens”.
Point 10. Figure 2: I am confused about what is shown in panel “d”. The body test suggests that this is the control inocula grown and sporulated on medium lacking hygromycin, and that is what the image looks like to me. However, The figure legend suggests that the medium contains 25 µg/mL hygromycin B. Please clarify.
Response 10: we are very grateful for your quite academic comment. We have made a serious mistake in the order of the panel “b”, panel “c” and panel “d”. we have revised the Figure 2 as follows.
Point 11. Line 244: Should the “106” here instead be 10^6, that is, with the “6” superscripted, equivalent to 1 million?
Response 11: Thank you for your suggestion, here 10^6 has been corrected as 106 in Line 244.
Point 12. Figure 4: the legend does not specify for panel “c” what gene was amplified. Presumably this is StcU-2?
Response 12: we sincerely thank you for your strict comment. we have added the information of panel “c” in legend of Figure4. “(c) Bands of wild strain, eight stably selected Cas9 gene edited StcU-2 mutants of A. apis and negative control for StcU-2 gene being edited”.
Point 13. I do not think the sequence of the EGFP gene shown in Figure 5 is worth including as a main figure. Obviously EGFP is present, or else the cells would not be fluorescent; the sequence of the gene provides no new information here.
Response 13: Thank you for your strict comments, we have deleted Figure 5 accordingly.
Point 14. Line 299: “Due to larger targeted gene section deletion events in some mutants, there were no PCR amplified bands compared to the wild A. apis strain.” What PCR product did the authors sequence to generate the data shown in Figure 6? The figure legend should specify this information. I am unsure what the absence of amplified bands refers to. In Figure 4c, there seemed to be PCR products amplified in all of the mutants. Assuming that panel represents PCR reactions targeting StcU-2, how do the authors explain the lack of PCR amplified bands that they refer to?
Response 14: Dear reviewer, we thank you for your deeper technical comments. To get the status of target gene, Cas-9, edition, 613bp target gene site bounding primers were used. The two-lane AA-StcU-2-1911-5 and AA-StcU-2-1911-7 have lost most of the gene sections which is due to major deletion effects. Though the result showed major deletion effects for the sec of easiness, we only used Protospacer and PAM sequences. Otherwise, in the wider sequence, there are other parts of sequences detected. The figure legends of Figure 4 and 6 have been revised as follows:
Figure 4. Different amplified StcU-2 gene edited Ascosphaera apis (A. apis) target gene bands. (a) Bands of wild strain, eight stably selected Cas9 genes edited StcU-2 mutants of A. apis and negative control for EGFP gene integration. (b) Bands of wild train and eight stably selected Cas9 genes edited StcU-2 mutants of A. apis and negative control for hph gene integration. (c) Bands of wild strain, eight stably selected Cas9 genes edited StcU-2 mutants of A. apis and negative control for StcU-2 gene being edited *For all: M denotes the DNA Marker 700 ladder; WT denotes A. apis wild strain. Lanes 1–8 denotes different independent StcU-2 transformed mutants; AAStcU-2-1911-1, AAStcU-2-1911-2, AAStcU-2-1911-3, AAStcU-2-1911-4, AAStcU-2-1911-5, AAStcU-2-1911-6, AAStcU-2-1911-7, and AAStcU-2-1911-8. –Ve denotes the negative control, which is a mix of sterile, double-distilled water and PCR without a template.
Figure 6. Clustal W sequence alignment of StcU-2 gene edited mutants. AA Wild Strain denotes Ascosphaera apis wild strain protospacer sequence and PAM sequence. Lanes AAStcU-2-1911-1, AAStcU-2-1911-2, AAStcU-2-1911-3 AAStcU-2-1911-4, AAStcU-2-1911-5, AAStcU-2-1911-7, and AAStcU-2-1911-8 denotes different independent StcU-2 gene edited mutant protospacer sequences and PAM sequence site editing.
Point 15. The panels in Figure 7 do not match the figure legend.
Response 15: Thank you for your strict comment. We have revised the Figure 7 legend based on your suggestion.
Point 16. Line 390: “The self-replicating plasmids were lost once the selection pressure was removed.” What is the evidence that this is the case?
Response 16: Thank you very much for your quite academic comments. we have deleted the sentence of “The self-replicating plasmids were lost once the selection pressure was removed. Only the modification at target gene locus of interest remained in the genome of the mutants” because the expression is not strict, and it is not necessary to be discussed according to the context of that discussion section.
Response to Reviewer 2 Comments:
Reviewer #2:
Point 1. Abstract: It should include background, objective, materials and methods, results, and a brief conclusion.
Response 1: Thank you for your comments. The abstract has been prepared in terms of background, objective, materials and methods, results, conclusion, and contribution based on the instruction of the journal and your suggestion.
Point 2. Line 18: successfully.
Response 2: Thank you very much. Corrected as suggested in Line 18.
Point 3. Line 23: a fungal honeybee brood pathogen.
Response 3: Thank you very much. “fungal honeybee brood pathogen” has been revised as “a fungal honeybee brood pathogen” in Line 23.
Introduction:
Point 4. Introduction needs more updated references.
Response 4: Thank you for your comments. We have updated the references of [4], [5], [6], [7,] [9], [13], [20], [21], [22], [23], [24], [25], [27], [28], [29], [31], [32], [33], [36], [41].
Point 5. Line 32: Ascosphaera apis
Response 5: Thank you for your suggestion. We have corrected as your suggestion in Line 32.
Point 6. Line 33: is a common.
Response 6: Thank you very much. “commone” has been corrected as “commom” in Line 33.
Point 7. Line 40: The DNA
Response 7: Thank you for your suggestion. We have corrected “DNA” as “The DNA” in Line 40.
Results
Point 8. Line 240: Ascosphaera apis
Response 8: Thank you very much. “A. apis” has been revised as “Ascosphaera apis” in Line 240.
Point 9. Line 334: Ascosphaera apis
Response 9: Thank you very much. “A. apis” has been revised as “Ascosphaera apis”.
Discussion
Point 10. Line 375: Ascosphaera apis
Response 10: Thank you very much. Revised as suggested.
Point 11. Discussion needs more improvements.
Response 11: Thank you for your comments. We have updated the discussion section as follows, and we believe that the quality has been much more improve and might satisfy you.
- Discussion
Fungi are common pathogens of insects. A. apis, causes chalkbrood disease and is a major pathogen of the honey bee. Its incidence and severity are increasing due to global climate change [11, 54]. Many treatments, including fungicides [55], plant essential oils [56], and biological strategies [13, 57], have been used to control chalkbrood disease. However, some of these approaches are not healthy for honey bee and contaminate the wax and honey. There are no effective measures to reliably control chalkbrood. Exploring effective and bee-friendly chalkbrood controlling approaches remain significant for apiculture and the environment. Spores are the primary infective agents of chalkbrood disease. Therefore, interference with spore production may help to control the disease. We used CRISPR/Cas9 gene editing approaches, and found that the StcU-2 gene functions in the sporulation and melanin biosynthesis in A. apis. Mutations of StcU-2 gene have substantial effects on spore-cyst production and can greatly reduce spore production. These results indicate that interfering with StcU-2 gene expression can interrupt sporulation in A. apis and that CRISPR/Cas9 can be successfully used in A. apis gene editing. This study provides baseline data for genes engineered using the Cas9/sgRNA-mediated method in A. apis. The results increase understanding of the function and mechanisms of different A. apis genes.
4.1. Successful Mutation of the StcU-2 Gene Inhibits A. apis Sporulation and Melanin production, Which Provides an Effective Way to Control Diseases Caused by Ascosphaera
Although A. apis alone is rarely fatal for honey bee colonies, it can kill colonies that are also interacting with other fungi[58-60]. Additionally, Ascosphaera contains of 28 species, which are all specialized to infect bees or bee colonies. The susceptibility of honey bees and other economic bee species to Ascosphaera varies [61-63]. Outbreaks of chalkbrood in the alfalfa leafcutting bee (Megachile rotundata) caused by A. aggregata can cause great economic losses [64]. Hence, an effective way to prevent chalkbrood disease will benefit for A. apis and other Ascosphaera controlling. The mechanism used by A. apis to be a successful entomopathogenic fungus depends on invasion and colonization strategies based on the production of an excessive number of spores. Spores germinate and survive after being ingested by larvae [65]. Cadavers of infected hosts are used to optimize production and dispersion of asco-spores [66], which are the unique infective stage [50]. To cause pathogenicity, the population of asco-spores in the gut of the host should be 5x105 asco-spores/larva [3] and kill the host (larvae) [66]. Thus, a reduction in the quality and number of spores of A. apis will be related to reduced spread and consequent reduction of A. apis pathogenicity to managed honey bees. In this study, A. apis mutants of the StcU-2 gene caused effective spore-cyst production lost and inhibited spore production in laboratory culture. This results in line with the decreasing the spore formation reduces the infection of Verticillium dahliae [67], and Aspergillus species [68].
Melanin of some fungi are spore extracellular matrices or part of the spore wall, it can help the pathogenic fungi invasion, trick the host recognizing antigen and block their immune system [69, 70]. In A. apis, melanin compacted as an electrodense layer throughout the entire cell wall of spore, and provided the necessary force to promote mycelium penetrate the larvae gut wall barrier during invasion process. [71]. Additionally, A. apis melanin maybe helps spores survive under adverse environments, and enable A. apis to survive and keep infectious after mummy larvae have been dead for a long time [72, 73]. In this study, all stable A. apis StcU-2 mutants grown from single protoplasts showed the expected phenotype of the colonies. They had albino spore-cysts that lacked melanin production, indicating that successive inactivation of the StcU-2 gene was highly related to melanin biosynthesis. Therefore, the successful inhibition of melanin of A. apis via interrupting StcU-2 gene will reduce the A. apis pathogenicity.
4.2. CRISPR/Cas9 Can be Used for A. apis Gene Function Studies
The recent advancement of sequencing technology has revealed a number of previously unknown gene of A. apis [16, 74]. Characterizations of these cryptic gene function in A. apis were greatly hampered due to the lack of handy gene manipulation methods. CRISPR/Cas9-mediated gene editing is a powerful tool for uncovering the gene functions [75]. However, there is no record for CRISPR/Cas9 used in A. apis. A. apis targeted gene disruptions using CRISPR/Cas9 nuclease at a specific genomic location induced effective DNA double strand breaks, which were then repaired via the error-prone NHEJ DNA repair pathway. This resulted in insertions and/or deletions (indels) of nucleotides, which then changed gene function [76]. An early exon DSB would easily be repaired via NHEJ repair, causing indels, or resulting in a frame shift, which will ultimately lead to a premature stop codon, or it will most likely lead to no protein product being translated at all [77]. Even though a truncated protein is produced, it will be non-functional as the sequence is altered at an early stage [78]. Thus, the first coding exon region intended sgRNA of A. apis produced successful cleavage of the StcU-2 gene with varied nucleotide indels and phenotypic and functional changes in sporulation. These results demonstrated that CRISPR/Cas9-mediated gene editing of the A. apis genome shows the potential of using CRISPR/Cas9 in different A. apis gene function studies. In this study we edited components of molecular pathways likely to contribute to melanin biosynthesis (sporulation) pathways. In comparison to traditional gene deletion approaches, the CRISPR/Cas9 mutagenesis approach for A. apis was advantageous, as the system remained active and successively mutagenized the target genes. Hence, the method provides the opportunity to retransform the cured strain using the same selection marker to obtain double-mutant strains.
4.3. Minimum Growth Limiting Level of Hygromycin B Antibiotic to Wild Strain A. apis Colonies Was Determined
The minimum growth limiting level of hygromycin B antibiotic to wild strain A. apis colonies, which was critical for transformant colonies selection after CRISPR/Cas9 gene editing, was 25 µg/mL. The previous hygromycin B concentration level in PDA media was suggested to be 50 µg/mL [35, 50], but there was no spore or mycelial growth was observed in the wild strain A. apis in referred media in this study. The protocol helps to identify A. apis transformants from non-transformed colonies. Hygromycin B resistant A. apis colonies can easily be identified by their ability to grow on PIDA plates containing 25 µg/mL hygromycin B. At hygromycin B levels below 25 µg/mL, some colonies have restricted growth, but at 25 µg/mL, all wild strain A. apis colonies were completely inhibited and regressed. This level was used for CRISPR/Cas9 transformants, which incorporated the hph gene in Cas9 enzyme carrier all-in-one plasmids. The contradiction with previous studies regarding hygromycin B [50], might be the level of PDA media temperature during introduction of the hygromycin B antibiotic, which could denature the antibiotic. Though there are thermo-resistant hygromycin B antibiotic strains that resist denaturing above 60°C [79], ordinary hygromycin B antibiotic denatures at temperatures above 50°C [80]. In this study, the hygromycin B antibiotic was added when the internal medium temperature was 48°C.
Point 12. Discussion needs more updated references.
Response 12: Thank you for your comments. we have updated the references of [59], [60], [63], [67], [68], [69], [70],[71], [73], [74], [75].
Conclusion
Point 13. Long and not clear conclusion, please a brief and clear it.
Response 13: Thank you for your comments. We have revised the conclusion as “We first time demonstrated effective CRISPR/Cas9 mediated StcU-2 gene editing in A. apis, and confirmed that the StcU-2 gene was related to sporulation and melanin biosynthesis. This result provides a promising way to reduce pathogenicity and the spread of A. apis by CRISPR/Cas9.”
Point 14. The authors have used honeybee and honey bee, please use one of them I suggest honey bee throughout the manuscript.
Response 14: Thank you for your suggestion. “honeybee” has been revised as “honey bee” throughout the manuscript as you suggested.

Reviewer 2 Report
Dear Dr.
Editor of Microorganisms
Authors Thank you very much for choosing me for reviewing to your esteemed Journal (Microorganisms).
Please find the comments.
Abstract: It should include background, objective, materials and methods, results, and a brief conclusion.
Line 18: successfully.
Line 23: a fungal honeybee brood pathogen.
Introduction:
· Introduction needs more updated references.
Line 32: Ascosphaera apis
Line 33: is a common.
Line 40: The DNA
Results
· Line 240: Ascosphaera apis
· Line 334: Ascosphaera apis
Discussion
· Line 375: Ascosphaera apis
· Discussion needs more improvements.
· Discussion needs more updated references.
Conclusion
· Long and not clear conclusion, please a brief and clear it.
-The authors have used honeybee and honey bee, please use one of them I suggest honey bee throughout the manuscript.
Author Response
Response to Reviewer 2 Comments:
Reviewer #2:
Point 1. Abstract: It should include background, objective, materials and methods, results, and a brief conclusion.
Response 1: Thank you for your comments. The abstract has been prepared in terms of background, objective, materials and methods, results, conclusion, and contribution based on the instruction of the journal and your suggestion.
Point 2. Line 18: successfully.
Response 2: Thank you very much. Corrected as suggested in Line 18.
Point 3. Line 23: a fungal honeybee brood pathogen.
Response 3: Thank you very much. “fungal honeybee brood pathogen” has been revised as “a fungal honeybee brood pathogen” in Line 23.
Introduction:
Point 4. Introduction needs more updated references.
Response 4: Thank you for your comments. We have updated the references of [4], [5], [6], [7,] [9], [13], [20], [21], [22], [23], [24], [25], [27], [28], [29], [31], [32], [33], [36], [41].
Point 5. Line 32: Ascosphaera apis
Response 5: Thank you for your suggestion. We have corrected as your suggestion in Line 32.
Point 6. Line 33: is a common.
Response 6: Thank you very much. “commone” has been corrected as “commom” in Line 33.
Point 7. Line 40: The DNA
Response 7: Thank you for your suggestion. We have corrected “DNA” as “The DNA” in Line 40.
Results
Point 8. Line 240: Ascosphaera apis
Response 8: Thank you very much. “A. apis” has been revised as “Ascosphaera apis” in Line 240.
Point 9. Line 334: Ascosphaera apis
Response 9: Thank you very much. “A. apis” has been revised as “Ascosphaera apis”.
Discussion
Point 10. Line 375: Ascosphaera apis
Response 10: Thank you very much. Revised as suggested.
Point 11. Discussion needs more improvements.
Response 11: Thank you for your comments. We have updated the discussion section as follows, and we believe that the quality has been much more improve and might satisfy you.
- Discussion
Fungi are common pathogens of insects. A. apis, causes chalkbrood disease and is a major pathogen of the honey bee. Its incidence and severity are increasing due to global climate change [11, 54]. Many treatments, including fungicides [55], plant essential oils [56], and biological strategies [13, 57], have been used to control chalkbrood disease. However, some of these approaches are not healthy for honey bee and contaminate the wax and honey. There are no effective measures to reliably control chalkbrood. Exploring effective and bee-friendly chalkbrood controlling approaches remain significant for apiculture and the environment. Spores are the primary infective agents of chalkbrood disease. Therefore, interference with spore production may help to control the disease. We used CRISPR/Cas9 gene editing approaches, and found that the StcU-2 gene functions in the sporulation and melanin biosynthesis in A. apis. Mutations of StcU-2 gene have substantial effects on spore-cyst production and can greatly reduce spore production. These results indicate that interfering with StcU-2 gene expression can interrupt sporulation in A. apis and that CRISPR/Cas9 can be successfully used in A. apis gene editing. This study provides baseline data for genes engineered using the Cas9/sgRNA-mediated method in A. apis. The results increase understanding of the function and mechanisms of different A. apis genes.
4.1. Successful Mutation of the StcU-2 Gene Inhibits A. apis Sporulation and Melanin production, Which Provides an Effective Way to Control Diseases Caused by Ascosphaera
Although A. apis alone is rarely fatal for honey bee colonies, it can kill colonies that are also interacting with other fungi[58-60]. Additionally, Ascosphaera contains of 28 species, which are all specialized to infect bees or bee colonies. The susceptibility of honey bees and other economic bee species to Ascosphaera varies [61-63]. Outbreaks of chalkbrood in the alfalfa leafcutting bee (Megachile rotundata) caused by A. aggregata can cause great economic losses [64]. Hence, an effective way to prevent chalkbrood disease will benefit for A. apis and other Ascosphaera controlling. The mechanism used by A. apis to be a successful entomopathogenic fungus depends on invasion and colonization strategies based on the production of an excessive number of spores. Spores germinate and survive after being ingested by larvae [65]. Cadavers of infected hosts are used to optimize production and dispersion of asco-spores [66], which are the unique infective stage [50]. To cause pathogenicity, the population of asco-spores in the gut of the host should be 5x105 asco-spores/larva [3] and kill the host (larvae) [66]. Thus, a reduction in the quality and number of spores of A. apis will be related to reduced spread and consequent reduction of A. apis pathogenicity to managed honey bees. In this study, A. apis mutants of the StcU-2 gene caused effective spore-cyst production lost and inhibited spore production in laboratory culture. This results in line with the decreasing the spore formation reduces the infection of Verticillium dahliae [67], and Aspergillus species [68].
Melanin of some fungi are spore extracellular matrices or part of the spore wall, it can help the pathogenic fungi invasion, trick the host recognizing antigen and block their immune system [69, 70]. In A. apis, melanin compacted as an electrodense layer throughout the entire cell wall of spore, and provided the necessary force to promote mycelium penetrate the larvae gut wall barrier during invasion process. [71]. Additionally, A. apis melanin maybe helps spores survive under adverse environments, and enable A. apis to survive and keep infectious after mummy larvae have been dead for a long time [72, 73]. In this study, all stable A. apis StcU-2 mutants grown from single protoplasts showed the expected phenotype of the colonies. They had albino spore-cysts that lacked melanin production, indicating that successive inactivation of the StcU-2 gene was highly related to melanin biosynthesis. Therefore, the successful inhibition of melanin of A. apis via interrupting StcU-2 gene will reduce the A. apis pathogenicity.
4.2. CRISPR/Cas9 Can be Used for A. apis Gene Function Studies
The recent advancement of sequencing technology has revealed a number of previously unknown gene of A. apis [16, 74]. Characterizations of these cryptic gene function in A. apis were greatly hampered due to the lack of handy gene manipulation methods. CRISPR/Cas9-mediated gene editing is a powerful tool for uncovering the gene functions [75]. However, there is no record for CRISPR/Cas9 used in A. apis. A. apis targeted gene disruptions using CRISPR/Cas9 nuclease at a specific genomic location induced effective DNA double strand breaks, which were then repaired via the error-prone NHEJ DNA repair pathway. This resulted in insertions and/or deletions (indels) of nucleotides, which then changed gene function [76]. An early exon DSB would easily be repaired via NHEJ repair, causing indels, or resulting in a frame shift, which will ultimately lead to a premature stop codon, or it will most likely lead to no protein product being translated at all [77]. Even though a truncated protein is produced, it will be non-functional as the sequence is altered at an early stage [78]. Thus, the first coding exon region intended sgRNA of A. apis produced successful cleavage of the StcU-2 gene with varied nucleotide indels and phenotypic and functional changes in sporulation. These results demonstrated that CRISPR/Cas9-mediated gene editing of the A. apis genome shows the potential of using CRISPR/Cas9 in different A. apis gene function studies. In this study we edited components of molecular pathways likely to contribute to melanin biosynthesis (sporulation) pathways. In comparison to traditional gene deletion approaches, the CRISPR/Cas9 mutagenesis approach for A. apis was advantageous, as the system remained active and successively mutagenized the target genes. Hence, the method provides the opportunity to retransform the cured strain using the same selection marker to obtain double-mutant strains.
4.3. Minimum Growth Limiting Level of Hygromycin B Antibiotic to Wild Strain A. apis Colonies Was Determined
The minimum growth limiting level of hygromycin B antibiotic to wild strain A. apis colonies, which was critical for transformant colonies selection after CRISPR/Cas9 gene editing, was 25 µg/mL. The previous hygromycin B concentration level in PDA media was suggested to be 50 µg/mL [35, 50], but there was no spore or mycelial growth was observed in the wild strain A. apis in referred media in this study. The protocol helps to identify A. apis transformants from non-transformed colonies. Hygromycin B resistant A. apis colonies can easily be identified by their ability to grow on PIDA plates containing 25 µg/mL hygromycin B. At hygromycin B levels below 25 µg/mL, some colonies have restricted growth, but at 25 µg/mL, all wild strain A. apis colonies were completely inhibited and regressed. This level was used for CRISPR/Cas9 transformants, which incorporated the hph gene in Cas9 enzyme carrier all-in-one plasmids. The contradiction with previous studies regarding hygromycin B [50], might be the level of PDA media temperature during introduction of the hygromycin B antibiotic, which could denature the antibiotic. Though there are thermo-resistant hygromycin B antibiotic strains that resist denaturing above 60°C [79], ordinary hygromycin B antibiotic denatures at temperatures above 50°C [80]. In this study, the hygromycin B antibiotic was added when the internal medium temperature was 48°C.
Point 12. Discussion needs more updated references.
Response 12: Thank you for your comments. we have updated the references of [59], [60], [63], [67], [68], [69], [70],[71], [73], [74], [75].
Conclusion
Point 13. Long and not clear conclusion, please a brief and clear it.
Response 13: Thank you for your comments. We have revised the conclusion as “We first time demonstrated effective CRISPR/Cas9 mediated StcU-2 gene editing in A. apis, and confirmed that the StcU-2 gene was related to sporulation and melanin biosynthesis. This result provides a promising way to reduce pathogenicity and the spread of A. apis by CRISPR/Cas9.”
Point 14. The authors have used honeybee and honey bee, please use one of them I suggest honey bee throughout the manuscript.
Response 14: Thank you for your suggestion. “honeybee” has been revised as “honey bee” throughout the manuscript as you suggested.
Round 2
Reviewer 2 Report
The manuscript can be accepted